# Intron-containing RNA from the HIV-1 provirus activates type I interferon and inflammatory cytokines

Sean Matthew McCauley[1], Kyusik Kim [1], Anetta Nowosielska [1], Ann Dauphin[1], Leonid Yurkovetskiy [1], William Edward Diehl [1] & Jeremy Luban [1,2]

HIV-1-infected people who take drugs that suppress viremia to undetectable levels are protected from developing AIDS. Nonetheless, HIV-1 establishes proviruses in long-lived CD4[+] memory T cells, and perhaps other cell types, that preclude elimination of the virus even after years of continuous antiviral therapy. Here we show that the HIV-1 provirus activates innate immune signaling in isolated dendritic cells, macrophages, and CD4[+] T cells. Immune activation requires transcription from the HIV-1 provirus and expression of CRM1-dependent, Rev-dependent, RRE-containing, unspliced HIV-1 RNA. If *rev* is provided *in trans*, all HIV-1 coding sequences are dispensable for activation except those *cis*-acting sequences required for replication or splicing. Our results indicate that the complex, post-transcriptional regulation intrinsic to HIV-1 RNA is detected by the innate immune system as a danger signal, and that drugs which disrupt HIV-1 transcription or HIV-1 RNA metabolism would add qualitative benefit to current antiviral drug regimens.

[1] Program in Molecular Medicine, University of Massachusetts Medical School, Worcester, MA 01605, USA. [2] Department of Biochemistry and Molecular Pharmacology, University of Massachusetts Medical School, Worcester, MA 01605, USA. Correspondence and requests for materials should be addressed to J.L. (email: jeremy.luban@umassmed.edu)

There are currently over 40 antiretroviral drug formulations that block essential steps in the HIV-1 replication cycle, including reverse transcription, integration, and proteolytic processing of the viral polyproteins[1,2]. Although these treatments are effective at reducing viremia, HIV-1 establishes proviruses in long-lived CD4+ memory T cells, and perhaps other cell types[3–5], that preclude elimination of the virus even after years of continuous antiviral therapy[6]. As a result, HIV-1-infected individuals must maintain a strict regimen of antiviral drugs or risk viral rebound and development of HIV-1 drug resistance[6–9]. Furthermore, despite the drastic reduction in viral replication, these individuals sometimes exhibit chronic inflammation associated with heightened risk of cardiovascular pathology[10–12]. Though the majority of proviruses that persist during antiviral therapy are defective for the production of infectious virions[13], many proviruses are expressed, raising the possibility that the HIV-1 provirus, or its transcripts, contribute to ongoing inflammation[14].

HIV-1 exploits a number of host and viral factors to avoid detection by the innate immune system[15]. Innate immune detection of HIV-1 has been reported to occur via a variety of mechanisms. Early after encounter with a target cell, HIV-1 CA interaction with host TRIM5[16,17], or, independently, HIV-1 gp41[18], activates transforming growth factor beta-activated kinase 1 (TAK1)-dependent signaling pathways. Plasmacytoid dendritic cells (pDCs) readily detect HIV-1 RNA via TLR7 and produce copious levels of type I interferon[19–21]. HIV-1 genomic RNA has also been reported to activate RIG-I signaling[22,23], and HIV-1 cDNA can be detected by the cytoplasmic DNA sensor cGAS[24–26]. Sensing of HIV-1 infection has also been suggested to occur following integration, with critical determinants including HIV-1 capsid[27], post-integration cGAS signaling[25], and abortive HIV-1 RNA transcripts[28].

The determinants for innate immune detection of HIV-1 were investigated in detail here. To start, dendritic cells were challenged with a large panel of single-cycle HIV-1 vectors. The results show that high-level activation of innate immune gene expression in transduced cells requires integration, as well as transcription from the provirus. Furthermore, DC maturation is dependent upon Rev-CRM1-mediated export of intron-bearing, HIV-1 genomic RNA. Transduced macrophages and CD4+ T cells behave like transduced DCs. Primary HIV-1 clones in the context of complete, replication-competent HIV-1 also give similar results. The observations reported here demonstrate that the innate immune system detects HIV-1's unique modes of RNA processing and suggest that inhibitors of HIV-1 transcription or HIV-1 RNA metabolism would be beneficial additions to current retroviral therapy.

## Results

**Dendritic cells mature in response to challenge with HIV-1.** To determine if HIV-1 proviruses activate innate immune signaling, human blood cells were transduced with single-cycle vectors, either a full-length, single-cycle HIV-1 clone with a frameshift in *env* and eGFP in place of *nef* (HIV-1-GFP)[29], or a minimal 3-part lentivector encoding GFP (Fig. 1a, and Supplementary Table 1)[16]. Monocyte derived dendritic cells (DCs) were challenged initially since HIV-1 transduction of these specialized antigen-presenting cells activates innate immune signaling[23,24,27,30,31]. To increase the efficiency of provirus establishment, vectors were pseudo-typed with the vesicular stomatitis virus glycoprotein (VSV G) and delivered concurrently with virus-like particles (VLPs) bearing SIV_{MAC}251 Vpx (Fig. 1b)[16,32]. Transduction efficiency, as determined by flow cytometry for GFP-positive cells, was 30-60% (Fig. 1b and Supplementary Fig 1a), depending on the blood donor.

DCs matured in response to HIV-1-GFP transduction, as indicated by increased mean fluorescence intensity of co-stimulatory or activation molecules, including HLA-DR, CD80, CD86, CD40, CD83, CCR7, CD141, ISG15, MX1, and IFIT1[33] (Fig. 1b, c and Supplementary Fig. 1a, b). Identical results were obtained with full-length, single-cycle vectors generated from primary, transmitted/founder clones that were derived by single genome sequencing, HIV-1_{AD17}[34], HIV-1_{Z331M-TF}[35], and HIV-1_{ZM249M}[36], the first virus being clade B, the other two clade C (Fig. 1d and Supplementary Fig. 1c). A single cycle HIV-2 vector induced maturation, indicating that this innate response was not unique to HIV-1 (Fig. 1e and Supplementary Fig. 1d).

Maturation was evident among both GFP positive and negative cells (Fig. 1b), the latter likely resulting from activation *in trans* by type 1 IFN as others have shown[27,31]. To test this idea, naive DCs were incubated with filtered supernatant from autologous DCs that had been transduced previously with HIV-1-GFP. Supernatant from DCs transduced with HIV-1-GFP, but not with minimal lentivector, upregulated CD86 and ISG15 on the naive DCs (Fig. 1f and Supplementary Fig 1e). Maturation activity was evident at a 1:1000 dilution of supernatant to which nevirapine had been added to preclude carry-over of transduction-competent HIV-1-GFP.

DCs matured when HIV-1-GFP transduction efficiency was augmented with nucleosides[37], rather than with SIV VLPs, indicating that Vpx was not required for maturation (Fig. 1g and Supplementary Fig. 1f). DCs were then challenged with replication-competent HIV-1 bearing CCR5-tropic Env, either T cell-tropic or macrophage-tropic[38], with or without Vpx-VLPs (Fig. 1h and Supplementary Fig. 1g). The percent of cells transduced by vector bearing either Env increased with Vpx, though DC maturation was observed under all conditions, even among the very few DCs transduced by T cell-tropic *env* (see inset of Fig. 1h). These results indicate that neither VSV G, nor Vpx, nor high-titer infection, was required for DC maturation.

In response to transduction with HIV-1-GFP, steady-state *CXCL10*, *IFNB1*, and *IL15* mRNAs reached maximum levels at 48 h, increasing 31,000-, 92-, and 140-fold relative to mock-treated cells, respectively (Fig. 1i, j). Correspondingly, IFNα2, CCL7, IL-6, CXCL10, and TNFα proteins accumulated in the supernatant (Fig. 1k). In contrast to the results with HIV-1-GFP, there were no signs of maturation after transduction with the 3-part minimal lentivector (Fig. 1b, j, k and Supplementary Fig. 1a, b, c, d).

**DC maturation requires reverse transcription and integration.** To determine if early stages in the HIV-1 replication cycle were necessary for maturation, reverse transcription was inhibited by nevirapine (NVP) or the HIV-1 RT mutant D185K/D186L, and integration was inhibited with raltegravir (RALT) or the HIV-1 IN mutant D116A (Supplementary Tables 1 and 2), as previously described[39]. Each of these four conditions abrogated maturation, as indicated by cell surface CD86 (Fig. 2a, and Supplementary Fig. 2a) and steady-state *CXCL10* mRNA (Fig. 2b). When integration was inhibited, *CXCL10* mRNA increased in response to challenge with HIV-1-GFP, but levels were nearly 1,000 times lower than when integration was not blocked (Fig. 2b). As an additional control, DCs challenged with the well-characterized, non-infectious, protease active-site mutant D25A[40] expressed 10,000 times less CXCL10 mRNA than did cells transduced with HIV-1-GFP (Fig. 2b).

HIV-1 virion RNA and newly synthesized viral cDNA are reported to be detected by RIG-I and by cGAS, respectively[23,24]. Signal transduction downstream of both sensors requires TBK1 and IRF3. The TBK1 inhibitor BX795 (Supplementary Table 2) blocked DC maturation in response to cGAMP but had no effect on maturation after HIV-1-GFP transduction (Fig. 2c and

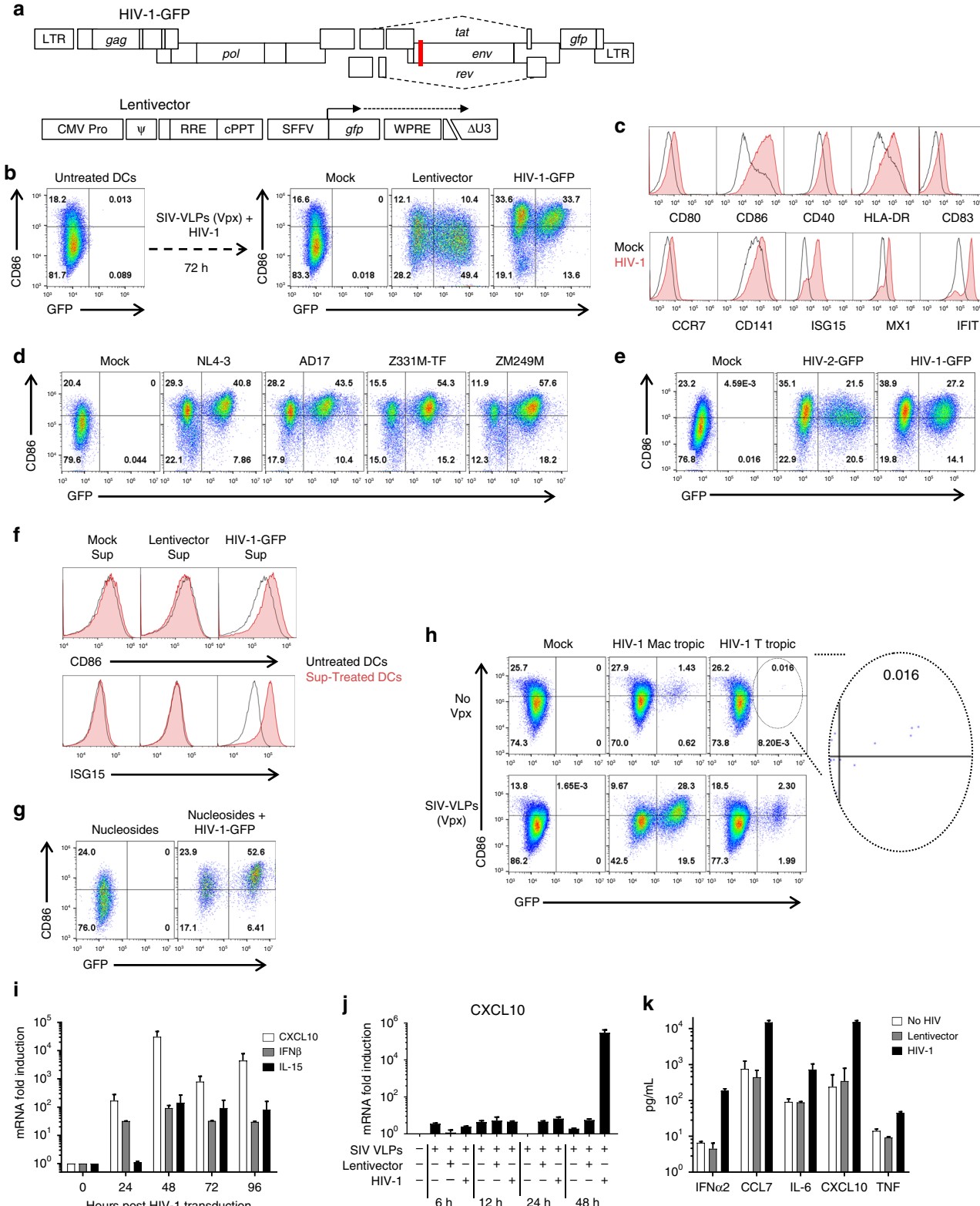

Supplementary Fig. 2b). Moreover, IRF3 knockdown (Supplementary Table 1)[16] suppressed activation of CD86 or ISG15 in response to cGAMP, but not in response to HIV-1 transduction (Fig. 2c and Supplementary Fig. 2b). Similarly, no effect on HIV-1-induced DC maturation was observed with knockdown of IRF1, 5, 7, or 9, or of STAT1 or 2, Supplementary Table 1)[16], or of pharmacologic inhibition of CypA, PKR, c-Raf, IkBa, NF-kB, MEK1 + 2, p38, JNK, Caspase 1, pan-Caspases, ASK1, eIF2a,

TBK1, IKKe, TAK1, or NLRP3 (Supplementary Table 2). Under the conditions used here, then, DC maturation required reverse transcription and integration but was independent of most well characterized innate immune signaling pathways.

**Provirus transcription is required for innate activation.** Completion of the HIV-1 integration reaction requires cellular DNA

**Fig. 1** HIV-1 transduction matures DCs. **a** Schematic of HIV-1-GFP, with frameshift in *env* (red line) and *gfp* in place of *nef*[29], and of the minimal lentivector, with self-inactivating ΔU3 LTR[42] and *gfp* driven by the SFFV promoter[16]. Unless indicated otherwise, vectors were pseudotyped with VSV G and cells were co-transduced with SIV$_{MAC}$251 VLPs bearing Vpx. **b** Flow cytometry of DCs for GFP and CD86, after treatment as indicated. **c** Flow cytometry histograms for the indicated markers 72 h after DC transduction with HIV-1 (red) or mock (black). **d** Flow cytometry of DCs for GFP and CD86 after transduction with single-cycle clones, HIV-1$_{NL4-3}$, HIV-1$_{AD17}$, HIV-1$_{Z331M-TF}$, or HIV-1$_{ZM249M}$. **e** Transduction of DCs with HIV-2$_{ROD}$-GFP, single-cycle vector. **f** Flow cytometry for CD86 and ISG15 of DCs treated for 24 h in the presence of nevirapine with a 1:1000 dilution of supernatant from autologous DCs transduced with the indicated vectors. **g** DC transduction with HIV-1-GFP in the absence of Vpx and the presence of 2 mM nucleosides. **h** 12-day spreading infection on DCs, with macrophage-tropic or T cell-tropic, replication-competent HIV-1, with or without SIV VLPs. **i** qRT-PCR quantitation of *CXCL10* (black), *IFNB1* (gray), or *IL15* (white) mRNAs from DCs transduced with HIV-1-GFP. **j** qRT-PCR quantitation of *CXCL10* mRNA in DCs transduced with either HIV-1-GFP or minimal lentivector, assessed at the indicated times post-transduction. **k** Cytokines in DC supernatant as assessed by luminex, 72 h after transduction with HIV-1-GFP (black) or minimal lentivector (gray). Shown are blood donor data representative of *n* = 12 (**b**), *n* = 4 (**c-k**), or *n* = 8 (**h**). To determine significance, the MFI of all live cells for each sample was calculated as fold-change versus control. The exception being **h** where the MFI of only GFP+ cells was compared. When data from each donor replicate within a experiment was combined, the difference in MFI for all experimental vs control conditions was significant in all cases, *p* < 0.01; one-way ANOVA, Dunnett's post-test. qRT-PCR and Luminex data were mean ± SD, *p* < 0.0001; two-way ANOVA, Dunnett's post-test

repair enzymes[41]. That DCs did not mature in response to transduction with minimal lentivectors (Fig. 1b, j, k) indicates that activation of the DNA repair process is not sufficient, and that transcription from the HIV-1-GFP provirus must be necessary for maturation. Indeed, *gag* expression from an integrated vector has been reported to be necessary for DC maturation[27]. To determine if any individual HIV-1 proteins were sufficient to mature DCs, a minimal lentivector was used to express codon optimized versions of each of the open reading frames possessed by HIV-1-GFP (Fig. 2d, Supplementary Fig. 2b, and Supplementary Table 1). Among these vectors was a *gag*-expression vector that produced as much p24 protein as did HIV-1-GFP (Fig. 2d). None of these vectors matured DCs (Fig. 2d and Supplementary Fig. 2d).

HIV-1-GFP was then mutated to determine if any protein coding sequences were necessary for DC maturation. For these and any subsequent experiments in which an essential viral component was disrupted within HIV-1-GFP, the factor in question was provided *in trans*, either during assembly in transfected HEK293 cells, or within transduced DCs, as appropriate (see Methods). Mutations that disrupted both *gag* and *pol*, either a double frameshift in *gag*, or a mutant in which the first 14 ATGs in *gag* were mutated, abolished synthesis of CA (p24) yet retained full maturation activity (Fig. 2e, Supplementary Fig. 2e, and Supplementary Table 1). Deletion mutations encompassing *gag*/*pol*, *vif*/*vpr*, *vpu*/*env*, or *nef*/U3-LTR, each designed so as to leave *cis*-acting RNA elements intact, all matured DCs (Fig. 2f, Supplementary Fig. 2e, and Supplementary Table 1). These results indicate that these HIV-1-GFP RNA sequences, as well as the proteins that they encode, were not required for DC maturation.

Tat and Rev coding sequences were individually disrupted by combining start codon point mutations with nonsense codons that were silent with respect to overlapping reading frames (Supplementary Table 1). Neither Δ*tat* nor Δ*rev* matured DCs upon transduction (Fig. 2g and Supplementary Fig. 2f). However, DCs matured upon co-transduction of Δ*tat* and Δ*rev*, or when minimal lentivectors expressing codon-optimized Tat and Rev were co-transduced *in trans* (Fig. 2g and Supplementary Fig. 2g). These results indicate that the maturation defect with the individual vectors was due to disruption of Tat and Rev function, and not due to a *cis*-acting defect of the mutant RNA.

The minimal 3-part lentivector expressed GFP from a heterologous promoter and had a deletion mutation encompassing the essential, *cis*-acting TATA box and enhancer elements[42], as well as in the *trans*-acting *tat* and *rev*, that inactivated the promoter in the proviral 5′ LTR (Fig. 1a). To test the importance of LTR-driven transcription for DC maturation by the HIV-1 provirus, the HIV-1 LTR was restored in the minimal vector (Fig. 2h, Supplementary Fig. 2g, and Supplementary Table 1); in

addition, GFP was inserted in place of *gag* as a marker for LTR expression, and the heterologous promoter was used to drive *tat*, *rev*, or both genes separated by P2A coding sequence (Fig. 2h). None of the LTR-driven, minimal vectors matured DCs (Fig. 2h and Supplementary Fig. 2g).

To determine if *tat* was necessary for DC maturation, *tat* and TAR were mutated in HIV-1-GFP and the LTR promoter was modified to be tetracycline-inducible, as previously described[43] (Tet-HIV-1 in Fig. 3a and Supplementary Table 1). The doxycycline-dependent reverse transactivator (*rtTA*) was delivered *in trans* by lentivector. In the presence of doxycycline (Supplementary Table 2), Tet-HIV-1 and rtTA matured DCs when given in combination, but neither vector matured DCs when given in isolation (Fig. 3a and Supplementary Fig. 3a). Additionally, the magnitude of cell surface CD86 was dependent on the doxycycline concentration, indicating that maturation was dependent on the level of HIV-1 transcription (Fig. 3b and Supplementary Fig. 3b). These results demonstrated that *tat* was not required for maturation, so long as the provirus was expressed.

**Rev-CRM1-mediated RNA export is necessary for activation**. To ascertain whether *rev* was necessary for DC maturation, the RTE from a murine intracisternal A-particle retroelement (IAP), and the CTE from SRV-1, were inserted in place of *nef* (HIV-RTE/CTE in Fig. 3c, Supplementary Fig. 3c, and Supplementary Table 1)[44]. Each of these elements utilizes the NXF1 nuclear RNA export pathway, thereby bypassing the need for CRM1 and *rev*[45]. p24 levels with this construct were similar to those of HIV-1-GFP, indicating that unspliced RNA was exported from the nucleus at least as well as with Rev (Fig. 3c). Nonetheless, the HIV-RTE/CTE vector did not mature DCs (Fig. 3c and Supplementary Fig. 3c), indicating that maturation was dependent upon *rev* and CRM1-mediated RNA export. Consistent with this conclusion, the CRM1 inhibitor leptomycin B (Supplementary Table 2) abrogated DC maturation by HIV-1-GFP (Fig. 3d and Supplementary Fig. 3d). In contrast, leptomycin B had no effect on DC maturation in response to Sendai virus infection (Fig. 3e and Supplementary Fig. 3d). ISG15 was used to monitor maturation in these experiments since, as previously reported for DCs, leptomycin B altered background levels of CD86[46].

**HIV-1 activates macrophages and CD4+ T cells**. To determine if innate immune detection of HIV-1 was unique to DCs, monocyte-derived macrophages and CD4+ T cells were examined. In response to transduction with HIV-1-GFP, macrophages upregulated CD86, ISG15, and HLA-DR, and CD4+ T cells upregulated MX1, IFIT1, and HLA-DR (Fig. 4a and Supplementary Fig. 4a-f). DCs, macrophages, and CD4+ T cells were then transduced side-by-side with mutant constructs to determine if the

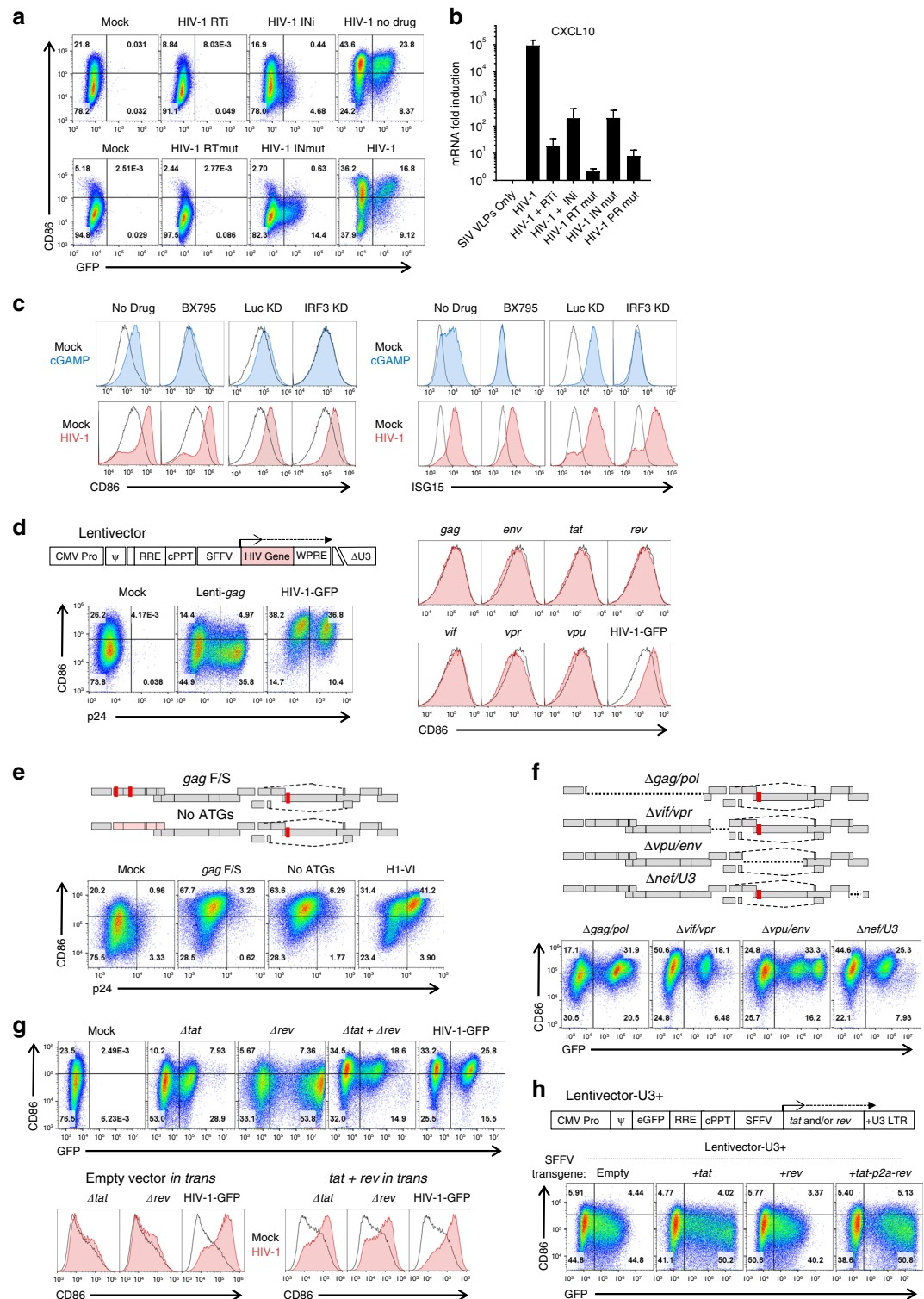

mechanism of innate immune activation was similar to that in DCs. As with DCs, HIV-1-GFP bearing the Δgag/pol deletion activated macrophages and CD4+ T cells (Fig. 4b and Supplementary Fig. 4a, d). Also in agreement with the DC results, neither the minimal lentivector, nor HIV-1-GFP bearing mutations in integrase, tat, or rev, matured any of the three cell types (Fig. 4b and Supplementary Fig. 4a, d).

CD4+ T cells were infected with either macrophage-tropic or T cell-tropic HIV-1 to determine whether replication-competent HIV-1 was similarly capable of innate immune activation in these cells, in the absence of VSV G. As with DCs, innate immune activation, as detected by MX1 and ISG15 upregulation, was observed in cells productively infected with HIV-1, but not with minimal lentivector (Fig. 4c and Supplementary Fig. 4g, h). Finally, to test the effect of HIV-1 proviral RNA on non-activated T cells, CD4+ T cells were co-transduced with Tet-HIV-1 and the rtTA3 vector, and cultured for 9 days in the absence of stimulation. Upon doxycycline treatment, T cells expressed GFP and MX1 (Fig. 4d and Supplementary Fig. 4i). As in DCs, dose-dependent activation was observed with doxycycline (Fig. 4d and

**Fig. 2** Native HIV-1 RNA regulation is necessary for DC maturation. **a** Assessment of GFP and CD86 by flow cytometry following transduction with, top, HIV-1-GFP in the presence of 5 μM nevirapine (RTi), 10 μM raltegravir (INi), or no drug, and, bottom, HIV-1-GFP bearing mutant RT-D185K/D186L (RTmut) or mutant IN-D116A (INmut). **b** qRT-PCR quantitation of *CXCL10* mRNA from the same DCs as **a**. **c** DCs treated with 1 μM of the TBK1 inhibitor BX795, or expressing shRNAs targeting either IRF3 or luciferase control[16], were challenged with 25 μg/mL cGAMP or HIV-1-GFP and assayed by flow cytometry for CD86 and ISG15. **d** Flow cytometry of DCs after transduction with minimal lentivectors expressing codon optimized HIV-1 genes; **e**, HIV-1-GFP in which translation was disrupted by two frameshifts in *gag* or by mutation of the first 14 AUGs in *gag*; **f**, HIV-1-GFP bearing deletion mutations encompassing *gag/pol*, *vif/vpr*, *vpu/env*, or *nef*/U3-LTR; **g**, HIV-1-GFP bearing mutations in *tat* or *rev*, co-transduced with both mutants, or co-transduced with minimal vector expressing *tat* and *rev* in trans; or **h** minimal lentivector with GFP in place of *gag*, SFFV promoter driving expression of *tat*, *rev*, or both, and repaired U3 in the 3′ LTR. When an essential viral component was disrupted within HIV-1-GFP, the factor in question was provided *in trans*, either during assembly in transfected HEK293 cells, or within transduced DCs, as appropriate (see Methods). Shown are blood donor data representative of *n* = 6 (**a**, **b**, **e**, **f**), *n* = 12 (**c**, **g**, **h**), *n* = 8 (**d**). To determine significance, the MFI of individual flow cytometry samples was calculated as fold-change versus control. When data from each donor replicate within a experiment were combined, the difference in MFI for all experimental vs control conditions was significant in all cases, *p* < 0.01; one-way ANOVA, Dunnett's post-test against HIV-1-GFP for **a**, **c**, **d**, **h** or lentivector control for **e**, **f**. qRT-PCR data are mean±SD (*p* < 0.0001; two-way ANOVA, Dunnett's post-test)

Supplementary Fig. 4i). These data indicate that innate immune activation by HIV-1, in all three cell types, requires integration, transcription, and Rev-dependent, HIV-1 intron-containing RNA.

## Discussion

The HIV-1 LTR generates a single primary transcript that gives rise to over 100 alternatively spliced RNAs[47]. The full-length, unspliced, intron-bearing transcript acts as viral genomic RNA in the virion and as mRNA for essential *gag*- and *pol*-encoded proteins. Expression of the unspliced transcript requires specialized viral and cellular machinery, HIV-1 Rev and CRM1[45], in order to escape from processing by the spliceosome. Results here indicate that unspliced or partially spliced HIV-1 RNA is detected by human cells as a danger signal, as has been reported for inefficiently spliced mRNAs from transposable elements in distantly related eukaryotes[48]. Transposable elements are mutagenic to the host genome and it stands to reason that molecular features such as transcripts bearing multiple, inefficient splice signals characteristic of retrotransposons, would activate innate immune signaling pathways.

HIV-1 genomic RNA contains extensive secondary and higher order structures that could be detected by innate immune sensors. Our knockdown of IRF3 and inhibition of TBK1, both required for signal transduction of the RNA sensors RIG-I, MDA5, and TLR3, did not impede HIV-1 maturation of DCs (Fig. 2c and Supplementary Fig. 2b). Furthermore, we suppressed an extensive list of innate signaling pathways and sensors including knockdowns of IRF's 1, 5, 7, and 9, STAT's 1 and 2, as well as pharmacologic inhibition of CypA, PKR, c-Raf, IkBa, NF-kB, MEK1 + 2, p38, JNK, Caspase 1, pan-Caspases, ASK1, eIF2a, IKKe, TAK1, or NLRP3 (Supplementary Fig. 2c and Supplementary Table 2). None of these perturbations had any effect on limiting innate immune activation by HIV-1, suggesting requirement for an alternative detection mechanism. Such mechanisms might include uncharacterized RNA sensors, more than one redundantly acting sensors, direct detection of stalled splicing machinery, or overload of the CRM1 nuclear export pathway itself.

The replication competent HIV-1 reservoir in memory CD4+ T cells has a 44 wk half-life and thus patients must take antiviral medication for life[49]. Long-lived, replication competent HIV-1 reservoirs in other cell types have not been clearly demonstrated, but these may also contribute to the HIV-1 reservoir[3]. The common genetic determinants in HIV-1 for maturation of CD4+ T cells, macrophages, and DCs suggests that HIV-1 is detected by a mechanism that is conserved across cell types, and that this mechanism would be active in any cell type that possesses a transcriptionally active provirus. Data here show that proviruses need not be replication competent to contribute to inflammation.

Rather, HIV-1 transcription and export of unspliced RNA, regardless of replication competence, is sufficient to induce immune activation. This almost certainly contributes to systemic inflammation during acute or untreated chronic infection. Whether inflammation is activated in response to provirus expression in cells of patients on antiretroviral therapy remains to be determined. Consistent with our findings, T cell activation correlates directly with the level of cell-associated HIV-1 RNA in patients receiving antiretroviral therapy[50]. Finally, our data suggests that new drugs that block HIV-1 transcription, Tat-mediated transcriptional elongation, or Rev-mediated preservation of unspliced transcripts[51], would limit inflammation, and offer an important addition to the current anti-HIV-1 drug armamentarium.

## Methods

**Data reporting**. No statistical methods were used to predetermine sample size. The experiments were not randomized. The investigators were not blinded to allocation during experiments and outcome assessment.

**Plasmids**. The plasmids used here were either previously described or generated using standard cloning methods[16]. The full list of plasmids used here, along with their purpose and characteristics, is provided in Supplementary Table 1. All plasmid DNAs with complete nucleotide sequence files are available at www.addgene.com.

**Cell culture**. Cells were cultured at 37 °C in 5% CO2 humidified incubators and monitored for mycoplasma contamination using the Lonza Mycoplasma Detection kit by Lonza (LT07-318). HEK293 cells (ATCC) were used for viral production and were maintained in DMEM supplemented with 10% FBS, 20 mM L-glutamine (ThermoFisher), 25 mM HEPES pH 7.2 (SigmaAldrich), 1 mM sodium pyruvate (ThermoFisher), and 1× MEM non-essential amino acids (ThermoFisher). Cytokine conditioned media was produced from HEK293 cells stably transduced with pAIP-hGMCSF-co (Addgene #74168), pAIP-hIL4-co (Addgene #74169), or pAIP-hIL2 (Addgene #90513), as previously described[16].

Leukopaks were obtained from anonymous, healthy, blood bank donors (New York Biologics). As per NIH guidelines (http://grants.nih.gov/grants/policy/hs/faqs_aps_definitions.htm), experiments with these cells were declared non-human subjects research by the UMMS IRB. PBMCs were isolated from leukopaks by gradient centrifugation on Histopaque-1077 (Sigma-Aldrich). CD14+ mononuclear cells were isolated via positive selection using anti-CD14 antibody microbeads (Miltenyi). Enrichment for CD14+ cells was routinely >98%.

To generate DCs or macrophages, CD14+ cells were plated at a density of 1 to 2 × 10^6 cells/mL in RPMI-1640 supplemented with 5% heat inactivated human AB+ serum (Omega Scientific, Tarzana, CA), 20 mM L-glutamine, 25 mM HEPES pH 7.2, 1 mM sodium pyruvate, and 1× MEM non-essential amino acids (RPMI-HS complete) in the presence of cytokines that promote differentiation. DCs were generated by culturing monocytes for 6 days in the presence of 1:100 cytokine-conditioned media containing human GM-CSF and human IL-4. DC preparations were consistently >99% DC-SIGN^high, CD11c^high, and CD14^low by flow cytometry. Macrophages were generated by culturing for 7 days with GM-CSF conditioned media in the absence of IL-4, and were routinely >99% CD11b. CD4+ T cells were isolated from PBMCs that had been depleted of CD14+ cells, as above, using anti-CD4 microbeads (Miltenyi), and were >99% CD4+. CD4+ T cells were then cultured in RPMI-1640 supplemented with 10% heat inactivated FBS, 20 mM L-

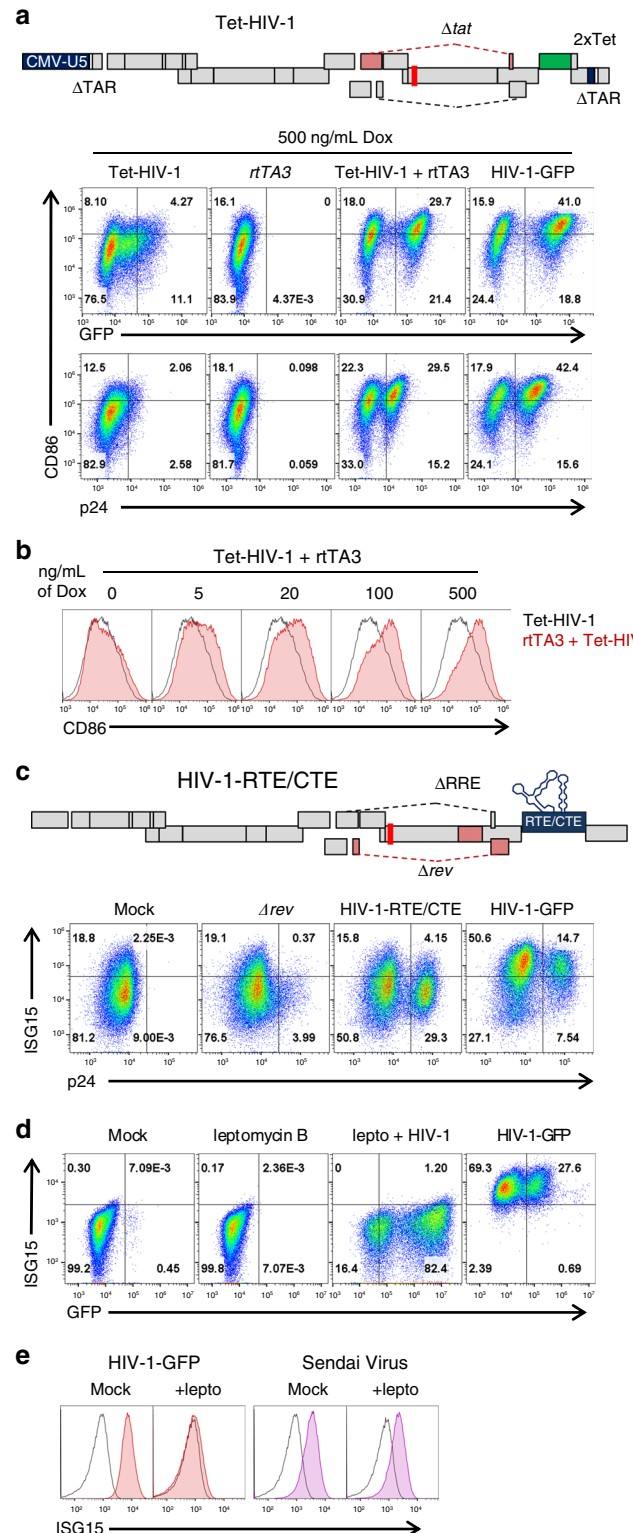

**Fig. 3** Rev-mediated RNA export is necessary for DC maturation but Tat is dispensable. **a** Optimized 2xTet operator[43] was cloned into the 3′LTR of HIV-1-GFPΔtat to generate Tet-HIV-1; the strand-transfer reactions that occur during reverse transcription generate a Tet-regulated 5′-LTR in the provirus. DCs transduced with Tet-HIV-1, rtTA3, or both, were treated for 3 d with 500 ng/mL doxycycline and assayed by flow cytometry for p24, GFP, and CD86. **b** DCs co-transduced with Tet-HIV-1 and rtTA3 were treated with increasing concentrations of doxycycline. **c** To generate HIV-1-RTE/CTE, the RTEm26CTE element[44] was cloned in place of *nef* in HIV-1-GFPΔ*rev*/ΔRRE. DCs were transduced with the indicated vectors and assessed for p24 and ISG15 by flow cytometry. **d** DCs were treated with 25 nM leptomycin B, transduced with HIV-1-GFP, and assessed for GFP and ISG15 by flow cytometry. **e** DCs were treated with 25 nM leptomycin B, transduced with HIV-1-GFP or infected with Sendai virus (SeV), and assessed for ISG15 by flow cytometry. Shown are blood donor data representative of $n = 10$ (**a**, **c**), $n = 4$ (**b**), $n = 6$ (**d**, **e**). To determine significance, the MFI of individual flow cytometry samples was calculated as fold-change versus control. When data from each donor replicate within a experiment was combined, the difference in MFI for all experimental vs control conditions was significant in all cases, $p < 0.01$; one-way ANOVA, Dunnett's post-test against dox negative control for **a**, **b** or HIV-1-GFP for **c**–**e**

transfected at a 7:1 ratio in terms of µgs of HIV-1 plasmid DNA to pMD2.G VSV G expression plasmid DNA[16]. Three-part lentivectors were produced by transfection of the lentivector genome, psPAX2 GagPol vector, and pMD2.G, at a DNA ratio of 4:3:1. These also include 2-part HIV-1-GFP constructs that are mutated in such a way as to prevent GagPol, Tat, or Rev production. As these would be defective for viral production, psPAX2 was included in the transfections at the same 4:3:1 ratio. VPX-bearing SIV-VLPs were produced by transfection at a 7:1 plasmid ratio of SIV3+ to pMD2.G[16]. Twelve hours after transfection, media was changed to the specific media for the cells that were to be transduced. Viral supernatant was harvested 2 days later, filtered through a 0.45 µm filter, and stored at 4 °C.

Virions in the transfection supernatant were quantified by a PCR-based assay for reverse transcriptase activity[16]. Five microliter transfection supernatant were lysed in 5 µL 0.25% Triton X-100, 50 mM KCl, 100 mM Tris-HCl pH 7.4, and 0.4 U/µL RNase inhibitor (RiboLock, ThermoFisher). Viral lysate was then diluted 1:100 in a buffer of 5 mM $(NH_4)_2SO_4$, 20 mM KCl, and 20 mM Tris–HCl pH 8.3. 10 µL was then added to a single-step, RT PCR assay with 35 nM MS2 RNA (IDT) as template, 500 nM of each primer (5′-TCCTGCTCAACTTCCTGTCGAG-3′ and 5′-CACAGGTCAAACCTCCTAGGAATG-3′), and hot-start Taq (Promega) in a buffer of 20 mM Tris-Cl pH 8.3, 5 mM $(NH_4)_2SO_4$, 20 mM KCl, 5 mM $MgCl_2$, 0.1 mg/mL BSA, 1/20,000 SYBR Green I (Invitrogen), and 200 µM dNTPs. The RT-PCR reaction was carried out in a Biorad CFX96 cycler with the following parameters: 42 °C 20 min, 95 °C 2 min, and 40 cycles [95 °C for 5 s, 60 °C 5 s, 72 °C for 15 s and acquisition at 80 °C for 5 s]. Two-part vectors typically yielded $10^7$ RT units/µL, and 3 part vector transfections yielded $10^6$ RT units/µL.

**Transductions**. $10^6$ DCs/mL, or $5 \times 10^5$ macrophages/mL, were plated into RPMI-HS complete with Vpx+ SIV-VLP transfection supernatant added at a dilution of 1:6. After 2 h, $10^8$ RT units of viral vector was added. In some cases, drugs were added to the culture media as specified in Supplementary Table 2. In most cases, transduced DC were harvested for analysis 3 days following challenge. For gene knockdown or for expression of factors *in trans*, $2 \times 10^6$ CD14+ monocytes/mL were transduced directly following magnetic bead isolation with 1:6 volume of SIV-VLPs and 1:6 volume of vector. When drug selection was required, 4 µg/mL puromycin was added 3 days after monocyte transduction and cells were selected for 3 days. SIV-VLPs were re-administered in all cases with HIV-1 or lentivector challenge. For DCs in Tet-HIV-1 experiments, fresh monocytes were SIV-VLP treated and co-transduced with rtTA3 and Tet-HIV-1. DCs were harvested 6 days later and treated with indicated doxycycline concentrations.

For deoxynucleoside-assisted transductions, DCs were plated at $10^6$ DCs/mL and treated with 2 mM of combined deoxynucleosides for 2 h before transduction with HIV-1. Deoxynucleosides were purchased from Sigma-Aldrich (2′ deoxyguanosine monohydrate, cat# D0901; thymidine, cat# T1895; 2′ deoxyadenosine monohydrate, cat# D8668; 2′deoxycytidine hydrochloride, cat# D0776). A 100 mM stock solution was prepared by dissolving each of the four nucleotides at 100 mM in RPMI 1640 by heating the medium at 80 °C for 15 min.

CD4+ T cells were stimulated in RPMI-FBS complete with 1:2000 hIL-2 conditioned media and 5 µg/mL PHA-P. After 3 days, T cells were replated at $10^6$ cells/mL in RPMI-FBS complete with hIL-2. Cells were transduced with $10^8$ RT units of viral vector per $10^6$ cells and assayed 3 days later. T cells were co-transduced with rtTA3 and Tet-HIV-1 every day for 3 days after PHA stimulation. Cells were then replated in RPMI-FBS complete with hIL-2. Transduced T cells

glutamine, 25 mM HEPES pH 7.2, 1 mM sodium pyruvate, 1× MEM non-essential amino acids (RPMI-FBS complete), and 1:2000 IL-2 conditioned media. Cells from particular donors were excluded from experiments if percent enrichment deviated more than 5% from the numbers mentioned above, or if there was no increase in activation markers in response to control stimuli (LPS, Sendai virus, and wild-type HIV-1-GFP).

**HIV-1 vector production**. HEK293E cells were seeded at 75% confluency in six-well plates and transfected with 6.25µL Transit LT1 lipid reagent (Mirus) in 250 µL Opti-MEM (Gibco) with 2.25 µg total plasmid DNA. 2-part HIV-1 vectors based on HIV-1-GFP[29] and described in detail in Supplementary Table 1 were

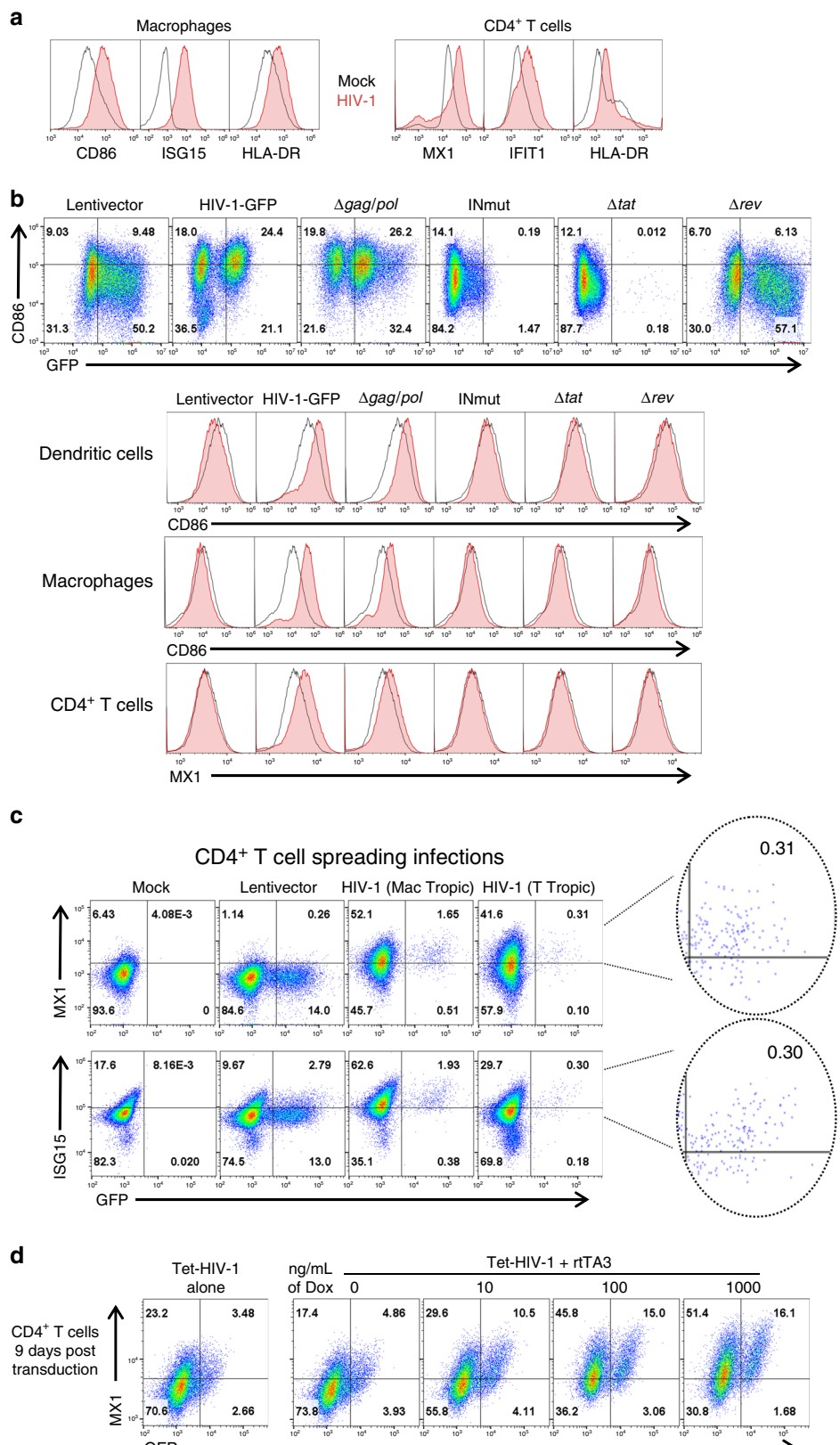

were cultured for 9 days with fresh media added at day 5. After 9 days, doxycycline was added at the indicated concentrations and assayed 3 days later.

**Non-HIV-1 challenge viruses**. Sendai Virus Cantell Strain was purchased from Charles River Laboratories. Infections were performed with 200 HA units/mL on DCs for 3 days before assay by flow cytometry.

**Spreading infections**. DCs were plated at 10^6 DCs/mL, in RPMI-HS complete media, with or without Vpx+ SIV-VLP transfection supernatant added at a dilution of 1:6. After 2 h, 10^8 RT units of HEK-293 transfection supernatant of either NL4-3-GFP-JRFL or NL4-3-GFP-JRCSF was added. Each is a construct of pNL4-3 in which *env* was replaced from the end of the signal peptide (therefore retaining the NL4-3 signal peptide and Vpu sequence) to the stop codon with either macrophage-tropic JR-FL *env* (GenBank: U63632.1) or T cell-tropic JR-CSF *env*

**Fig. 4** Innate immune activation in macrophages and CD4+ T cells. **a** Macrophages and CD4+ T cells were transduced with HIV-1-GFP and assayed 3 days later for the indicated activation markers. **b** DCs, macrophages, and CD4+ T cells were challenged with HIV-1-GFP or the indicated mutants. When an essential viral component was disrupted within HIV-1-GFP, the factor in question was provided *in trans* during assembly in transfected HEK293 cells, as appropriate (see Methods). The upper panel shows flow cytometry of the DCs for GFP and CD86. The histograms show CD86 for DCs and macrophages or MX1 for CD4+ T cells. **c** 12-day spreading infections on CD4+ T cells with either macrophage-tropic or T cell-tropic, replication-competent HIV-1. **d** CD4+ T cells were stimulated for 3 days with PHA and IL2, and transduced with Tet-HIV-1 and rtTA3. Cells were then cultured without stimulation for 9 days. Doxycycline was then added at the indicated concentrations. Cells were assayed for GFP and MX1 3 days later. Shown are blood donor data representative of $n = 6$ (**a**), $n = 4$ (**b**–**d**). To determine significance, the MFI of individual flow cytometry samples was calculated as fold-change versus control. The exception being **c** where the MFI of only GFP+ cells was compared. When data from each donor replicate within a experiment were combined, the difference in MFI for all experimental vs control conditions was significant in all cases, $p < 0.01$; one-way ANOVA, Dunnett's post-test against lentivector control for **a**–**c** or dox control for **d**

(GenBank: M38429.1). Every 3 days (for a total of 12 days) samples were harvested for detection of viral RT activity in supernatant and flow cytometry assessment.

CD4+ T cells were stimulated in RPMI-FBS complete with 1:2000 hIL-2 conditioned media and 5 µg/mL PHA-P. After 3 days, T cells were replated at $10^6$ cells/mL in RPMI-FBS complete with hIL-2 and transduced with $10^7$ RT units of NL4-3-GFP with JRFL (mac tropic) or JRCSF (T tropic) env. Cells were harvested every 3 days (for a total of 12 days) and assayed for infectivity and activation via flow cytometry.

**Supernatant transfer experiments.** Supernatant from HIV-1-GFP and minimal lentivector challenged DCs was harvested 3 days post transduction. This was centrifuged at $500 \times g$ for 5 mins and filtered through a 0.45 µm filter to remove cell debris. DCs of the same individual donors were then treated with the harvested supernatant at a dilution of 1:1000 (1 µL in 1 mL of culture). This was performed in the presence of nevirapine to inhibit secondary infection from remaining, cell free virions. Treatments of supernatants of HIV-1 challenged DCs were compared directly to unchallenged DC supernatant treatments.

**Cytokine analysis.** Supernatants from DCs were collected 3 days following transduction with HIV-1-GFP or minimal lentivector. Supernatant was spun at $500 \times g$ for 5 mins and filtered through a 0.45 µm filter. Multiplex soluble protein analysis was carried out by Eve Technologies (Calgary, AB, Canada).

**qRT-PCR.** Total RNA was isolated from $5 \times 10^5$ DCs using RNeasy Plus Mini (Qiagen) with Turbo DNase (ThermoFisher) treatment between washes. First-strand synthesis used Superscript III Vilo Master mix (Invitrogen) with random hexamers. qPCR was performed in 20 µL using 1× TaqMan Gene Expression Master Mix (Applied Biosystems), 1 µL cDNA, and 1 µL TaqMan Gene Expression Assays (ThermoFisher) specified in Supplementary Table 3. Amplification was on a CFX96 Real Time Thermal Cycler (Bio-Rad) using the following program: 95 °C for 10 min [45 cycles of 95 °C for 15 s and 60 °C for 60 s]. Housekeeping gene OAZ1 was used as control[16].

**Flow cytometry.** $10^5$ cells were surface stained in FACS buffer (PBS, 2% FBS, 0.1% Sodium Azide), using the antibodies in Supplementary Table 4. All antibodies were used at a dilution of 1:100 with the exception of the p24 antibody (KC57 Beckman-Coulter) which was used at a 1:200 dilution. Cells were then fixed in a 1:4 dilution of BD Fixation Buffer and assayed on a BD C6 Accuri. BD Biosciences Fixation and Permeabilization buffers were utilized for intracellular staining. Data was analyzed in FlowJo.

**Sampling.** All individual experiments were performed with biological duplicates, using cells isolated from two different blood donors. Flow cytometry plots in the figures show representative data taken from experiments performed with cells from the number of donors indicated in the figure legends.

**Statistical analysis.** Experimental $n$ values and information regarding specific statistical tests can be found in the figure legends. The mean fluorescence intensity for all live cells analyzed under a given condition was calculated as fold-change to negative control/mock. The exception to this methodology was in Figs. 1g and 4c where the percent infected cells was too low to use MFI for the bulk population; in these cases MFI was determined for the subset of cells within the GFP+ gate. Significance of flow cytometry data was determined via one-way ANOVA. A Dunnett's post-test for multiple comparisons was applied, where MFI fold change was compared to either mock treatment or positive treatment depending on the experimental question. qRT-PCR and luminex data was analyzed via two-way Anova, with Dunnett's post-test comparing all samples to mock. All ANOVAs were performed using PRISM 7.02 software (GraphPad Software, La Jolla, CA).

## Data availability

The plasmids described in Supplementary Table 1, along with their complete nucleotide sequences, are available at www.addgene.com. The data that support the findings of this study are available from the corresponding author upon reasonable request. A reporting summary for this article is available as a Supplementary Information file.

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

## Acknowledgements

We thank the 290 anonymous blood donors who contributed leukocytes to this project, and Ben Berkhout, Abraham Brass, Barbara Felber, Beatrice Hahn, Eric Hunter, Massimo Pizzato, and Didier Trono for reagents. This work was supported by NIH Grants 5R01AI111809, 5DP1DA034990, and 1R01AI117839, to J.L.

## Author contributions

S.M.M. and J.L. designed the experiments and S.M.M., K.K., A.D., A.N., L.Y., and W.E.D. conducted them. All authors analyzed the data and S.M.M. and J.L. wrote the manuscript.

## Additional information

**Competing interests:** The authors declare no competing interests.

