## [Peer Review File · Nature Communications]

Reviewers' comments:

Reviewer #1 (Remarks to the Author):

This article from McCauley et al identifies innate immune sensing of unspliced or minimally spliced HIV RNA following HIV infection as an important driver of DC, macrophage and Tcell maturation or activation. Interestingly, they show that many of the RNA encoding key structural genes were dispensible for these effects and that the determinants were intronic RNA, Tat and Rev activity including CRM1 dependent nuclear export.

The authors propose that the innate immune activation triggered in this manner may be responsible for the residual inflammation responsible for a host of non-AIDS morbidity in patients on ART, in particular cardiovascular morbidity. The experimental systems here are more likely to approximate productive infection than latent infection as seen with patients on long term ART. The authors should include a caveat that the levels of transcription that have been described in the in vivo setting of patients on ART, is very low and we cannot tell how much the effects described in the current studies apply. Nevertheless the observations (primarily in DC) in the setting most compatible with productive infection are novel and are of interest.

Specific issues:

While innate immune sensing of expressed RNA might be a contributor to inflammation and immune dysregulation during untreated infection in vivo, the authors should acknowledge that the experimental system used here is not proof that the low levels of residual HIV expression that have been described in HIV+ patients on treatment are sufficient to trigger comparable innate immune effects. This latter possibility is too heavily stressed in the introduction and in the concluding paragraph given the differences in expected RNA expression by latently infected cells and productively infected cells. The authors should be more circumspect in referencing the possibility that this innate immune sensing and signaling pathway is a primary driver of residual inflammation and immune dysregulation in patients on ART.

Figure 2B also shows that a protease (PR) mutant is impaired in its ability to elicit CXCL10 RNA. This is not discussed and would seem not to fit the paradigm that intronic RNA expression during relatively early stages of infection alone is responsible for the DC maturation effect. Can the authors comment on this?

(page 10 and Fig Supp 2c) Why does inhibition of CRM1 not affect the maturation effect since the CRM1 nuclear export pathway was shown to be necessary to trigger innate immune sensing ?

Is it possible that more than one pattern recognition receptor is responsible for the maturation effect and therefore knockdown of individual PRR signaling pathways is insufficient to exclude their contribution?

Reviewer #2 (Remarks to the Author):

The authors present evidence suggesting that cells have a mechanism by which they sense unspliced RNA in the cytoplasm. This is said to be innate immune sensing mechanism that acts as a danger signal to induce DC maturation and T cell ISG upregulation. The evidence is based on complex viral vectors in which various regions are mutated to prevent protein or RNA expression.

This is a sophisticated study in retrovirology and is quite thorough. The experiments include relevant controls and the paper is clear and concise.

The finding is quite unexpected, as there was no reason to think that HIV RNA is any different from any other RNA in the cell. There is no evidence for the existence of a cytoplasmic sensor of unspliced RNA and it difficult to imagine how such a sensor would work. It is also the case that cells express many cellular genes in alternatively spliced forms such that they are not fully spliced. The authors have ruled-out known RNA sensors such as RIG-I and MAVS but do not suggest other possible sensors. The model calls for yet another viral sensor, one that is not the same as the one predicted in the Nature paper from several ago from Manel et al. in which they claimed that viral capsid protein was sensed by a "cryptic sensor". The current paper would be much stronger if they had identified the sensor.

Another question regards the significance of the mechanism in vivo. DCs are not infected by HIV in vivo. Nor can this mechanism serve as the cause of inflammation in patients suppressed with antiretroviral drugs. There are only 1 in a million latently infected cells and the majority of these do not produce viral RNA and there is no evidence that latently infected cells make IFN. The authors show the effect in CD4 T cells and macrophages, although that data is much weaker. That data in Fig 4C is said to be a spreading infection. However, the number of infected cells is very low (0.3%) after 12 days, suggesting that the virus was not spreading. Moreover, while the GFP+ cells are MX1+ and ISG15+, the uninfected cells are also MX1+ and ISG15+. It could be that the MX1+ and ISG15+ cells are the ones that get infected and the virus doesn't induce their expression.

An unusual feature of the phenomenon in all of the experiments, is that the uninfected bystander cells act exactly as the infected cells. The authors state that this is due to the induction of IFN by the infected cells which then acts on the bystanders. That, however, is hard to believe. It would take time for IFN to accumulate in the medium such that the bystanders would become induced. In addition, the authors make this claim based on two publications from other labs that have not been substantiated. They show by RT-PCR some up-regulation of IFN mRNA but there is no measurement of secreted IFN to show that this is produced by the infected cells. That is an important question. The authors should measure secreted IFN and show that the concentration achieved in the culture is sufficient to mature the DCs.

Reviewers comments are itemized below in *italic 10 font*.

Authors responses to each specific comment are in bold blue 12 font, with text line numbers in red.

Reviewer #1 (Remarks to the Author)

This article from McCauley et al identifies innate immune sensing of unspliced or minimally spliced HIV RNA following HIV infection as an important driver of DC, macrophage and T cell maturation or activation. Interestingly, they show that many of the RNA encoding key structural genes were dispensable for these effects and that the determinants were intronic RNA, Tat and Rev activity including CRM1 dependent nuclear export. The authors propose that the innate immune activation triggered in this manner may be responsible for the residual inflammation responsible for a host of non-AIDS morbidity in patients on ART, in particular cardiovascular morbidity. The experimental systems here are more likely to approximate productive infection than latent infection as seen with patients on long term ART. The authors should include a caveat that the levels of transcription that have been described in the in vivo setting of patients on ART, is very low and we cannot tell how much the effects described in the current studies apply. Nevertheless the observations (primarily in DC) in the setting most compatible with productive infection are novel and are of interest.

Specific issues:

1. While innate immune sensing of expressed RNA might be a contributor to inflammation and immune dysregulation during untreated infection in vivo, the authors should acknowledge that the experimental system used here is not proof that the low levels of residual HIV expression that have been described in HIV+ patients on treatment are sufficient to trigger comparable innate immune effects. This latter possibility is too heavily stressed in the introduction and in the concluding paragraph given the differences in expected RNA expression by latently infected cells and productively infected cells. The authors should be more circumspect in referencing the possibility that this innate immune sensing and signaling pathway is a primary driver of residual inflammation and immune dysregulation in patients on ART.

We have modified the text accordingly on lines 26-27, 48-49, and 253 to 254.

2. Figure 2B also shows that a protease (PR) mutant is impaired in its ability to elicit CXCL10 RNA. This is not discussed and would seem not to fit the paradigm that intronic RNA expression during relatively early stages of infection alone is responsible for the DC maturation effect. Can the authors comment on this?

We thank the reviewer for pointing out that we had neglected to describe the results with the PR mutant. Text has now been added to lines 104-106. The PR mutant provided an additional negative control for the effect of non-infectious particles on DCs. PR mutant particles do not mature and replication is blocked at a very early step in virion entry. The PR mutant behaved exactly as our model would predict, inducing 10,000-times less CXCL10 mRNA than did cells transduced with HIV-1-GFP (Fig. 2b).

3. (page 10 and Fig Supp 2c) Why does inhibition of CRM1 not affect the maturation effect since the CRM1 nuclear export pathway was shown to be necessary to trigger innate immune sensing?

We thank the reviewer for alerting us to the fact that our text had mistakenly included CRM1 on a list of factors with no effect on HIV-1-induced DC maturation. Indeed, CRM1-inhibition blocks maturation. We have removed CRM1 from this text (now lines 235-236).

4. Is it possible that more than one pattern recognition receptor is responsible for the maturation effect and therefore knockdown of individual PRR signaling pathways is insufficient to exclude their contribution?

The reviewer is correct that more than one receptor may be necessary for maturation. On line 240 we now mention that more than one redundantly acting sensors might be required.

Reviewer #2 (Remarks to the Author):

The authors present evidence suggesting that cells have a mechanism by which they sense unspliced RNA in the cytoplasm. This is said to be innate immune sensing mechanism that acts as a danger signal to induce DC maturation and T cell ISG upregulation. The evidence is based on complex viral vectors in which various regions are mutated to prevent protein or RNA expression. This is a sophisticated study in retrovirology and is quite thorough. The experiments include relevant controls and the paper is clear and concise. The finding is quite unexpected, as there was no reason to think that HIV RNA is any different from any other RNA in the cell. There is no evidence for the existence of a cytoplasmic sensor of unspliced RNA and it difficult to imagine how such a sensor would work. It is also the case that cells express many cellular genes in alternatively spliced forms such that they are not fully spliced. The authors have ruled-out known RNA sensors such as RIG-I and MAVS but do not suggest other possible sensors. The model calls for yet another viral sensor, one that is not the same as the one predicted in the Nature paper from several ago from Manel et al. in which they claimed that viral capsid protein was sensed by a “cryptic sensor”. The current paper would be much stronger if they had identified the sensor.

1. Another question regards the significance of the mechanism in vivo. DCs are not infected by HIV in vivo. Nor can this mechanism serve as the cause of inflammation in patients suppressed with antiretroviral drugs. There are only 1 in a million latently infected cells and the majority of these do not produce viral RNA and there is no evidence that latently infected cells make IFN. The authors show the effect in CD4 T cells and macrophages, although that data is much weaker. That data in Fig 4C is said to be a spreading infection. However, the number of infected cells is very low (0.3%) after 12 days, suggesting that the virus was not spreading.

This experiment was performed at the lowest MOI possible in response to the explicit concern of one of the reviewers that our innate immune activation phenotype might be an artifact of non-physiologic, HIGH-titer, infection. By challenging cells with limiting quantities of virus, we showed that maturation occurs even at more physiologic levels of infection.

2. Moreover, while the GFP+ cells are MX1+ and ISG15+, the uninfected cells are also MX1+ and ISG15+. It could be that the MX1+ and ISG15+ cells are the ones that get infected and the virus doesn't induce their expression.

- As discussed in response to the reviewer's comment #4 below, clear explanation for maturation of the uninfected cells is offered by our new Fig. 1f.

- The hypothesis that MX1+/ISG15+ cells are selectively infected goes against years of innate immune research. Clear evidence against this hypothesis is that many of the vectors in our manuscript that do not induce maturation give higher titers than does WT HIV-1-GFP. For example, see Fig. 2g, where GFP+ cells are 41% for HIV-1-GFP and 61% for delta-rev.

3. An unusual feature of the phenomenon in all of the experiments, is that the uninfected bystander cells act exactly as the infected cells. The authors state that this is due to the induction of IFN by the infected cells which then acts on the bystanders. That, however, is hard to believe. It would take time for IFN to accumulate in the medium such that the bystanders would become induced.

Given the speed of JAK/STAT signaling, and given the 10,000-fold induction of IFN in our system (Fig. i), we thought it LIKELY that bystander cells would be activated. As discussed in response to the Reviewer's Comment #4, our new Fig. 1f shows that IFN in the culture of DCs challenged with HIV-1-GFP was at least 1,000-times the concentration necessary to mature DCs.

4. In addition, the authors make this claim based on two publications from other labs that have not been substantiated. They show by RT-PCR some up-regulation of IFN mRNA but there is no measurement of secreted IFN to show that this is produced by the infected cells. That is an important question. The authors should measure secreted IFN and show that the concentration achieved in the culture is sufficient to mature the DCs.

- The statement that there was no measurement of secreted IFN is not correct. Luminex data in Figure 1k had shown that HIV-1 induces IFN in the tissue culture supernatant.

- The suggestion that we assess maturation activity by IFN in the culture supernatant has greatly improved our manuscript. New Fig. 1f shows that the activity in culture supernatant from DCs challenged with HIV-1-GFP was at least 1,000-times what is required to mature DCs. To demonstrate this, naive autologous DCs were incubated in filtered culture supernatant, in the presence of an RT inhibitor. Consistent with our other experiments addressing the viral determinants for maturation, supernatant did not possess maturation activity if it was taken from DCs efficiently transduced with the minimal HIV-1 vector. This new experiment is discussed on **lines 73 to 78.**